# Cytokine Expression and Haptoglobin Levels in Bovine Fetuses Spontaneously Aborted by Intracellular Infectious Agents and by Probable Infectious Etiology

**DOI:** 10.3390/ani15192878

**Published:** 2025-10-01

**Authors:** Emiliano Sosa, Natalia Pla, Dadin Prando Moore, Juan Agustín García, Lucía María Campero, María Andrea Fiorentino, Evangelina Miqueo, Erika González Altamiranda, Fermín Lázaro, Karen Morán, María Guillermina Bilbao, Silvina Quintana, Maia Solange Marín, Germán José Cantón

**Affiliations:** 1Instituto de Innovación para la Producción Agropecuaria y el Desarrollo Sostenible (IPADS), Instituto Nacional de Tecnología Agropecuaria (INTA), Consejo Nacional de Investigaciones Científicas y Técnicas (CONICET), Estación Experimental Agropecuaria, Balcarce 7620, Argentina; sosa.emiliano@inta.gob.ar (E.S.); moore.dadin@inta.gob.ar (D.P.M.); garcia.juanagustin@inta.gob.ar (J.A.G.); campero.lucia@inta.gob.ar (L.M.C.); fiorentino.maria@inta.gob.ar (M.A.F.); galtamiranda.erika@inta.gob.ar (E.G.A.);; 2Facultad de Ciencias Exactas y Naturales, Universidad Nacional de Mar del Plata, Mar del Plata 7600, Argentina; nataliapla996@gmail.com (N.P.); marinmaia@yahoo.com.ar (M.S.M.); 3Facultad de Ciencias Agrarias, Universidad Nacional de Mar del Plata, Balcarce 7620, Argentina; miqueo.evangelina@inta.gob.ar; 4Facultad de Ciencias Veterinarias, Universidad Nacional de La Pampa, General Pico 6360, Argentina; morankaren89@gmail.com (K.M.); mgbilbao@vet.unlpam.edu.ar (M.G.B.); 5Instituto de Investigaciones en Producción, Sanidad y Ambiente (IIPROSAM), CONICET- Universidad Nacional de Mar del Plata, Mar del Plata 7600, Argentina; silvinaquintana.bm@gmail.com; 6CONICET, Mar del Plata 7600, Argentina

**Keywords:** bovine abortion, *Neospora caninum*, *Brucella abortus*, bovine viral diarrhea virus, immune response

## Abstract

Cattle abortions pose a serious challenge to livestock health and farm income, yet many cases lack a clear cause even when signs of infection are present. We examined immune activity in aborted bovine fetuses at mid and late gestation caused by three common agents that invade cells, comparing them with fetuses showing infection-like lesions without a known cause and with healthy controls. In mid-gestation losses by intracellular agents, we found a surge in proteins that drive inflammation, indicating a strong early defensive reaction. In contrast, late-pregnancy losses showed a drop in those inflammatory signals, suggesting a shift toward a more subdued immune state. Fetuses with unknown infections also showed distinct patterns. Although differences in haptoglobin levels were observed in mid-gestation fetuses aborted due to intracellular agents compared to controls, the overall concentration remained low and may be of limited biological relevance. Variations in the fetal immune response according to pathogen type and gestational stage provide novel insights into the pathogenesis of key intracellular agents, thereby strengthening diagnostic approaches and contributing to improved herd health and agricultural sustainability.

## 1. Introduction

Bovine abortion causes severe economic losses to the livestock industry [1,2]. Diagnosing bovine abortion is challenging, with a definitive etiological diagnosis reported in no more than 50% of analyzed aborted fetuses [1,3,4,5,6]. In Argentina, the most frequently identified intracellular infectious agents associated with bovine abortion are *Neospora caninum*, *Brucella abortus*, and Bovine Viral Diarrhea Virus (BVDV) [1,3,5]. *N. caninum* is a major cause of abortion in cattle worldwide, with a complex and only partially understood pathogenesis [7,8]. Diagnosis is based on the detection of the protozoan and the presence of characteristic histopathological changes [5,9,10]. Brucellosis is a zoonotic disease of significant concern for both animal health and the economy, with bacterial isolation and identification being the gold standard for diagnosis [11]. In Argentina, *B. abortus* was historically one of the most frequent causes of abortion, but since the implementation of a National Control and Eradication Plan in 1980, its incidence has decreased [1]. BVDV is endemic in the region and has a complex pathogenesis. Infection during gestation can result in transplacental transmission, leading to embryonic death, abortion, teratogenic effects, or the birth of persistently infected (PI) but clinically healthy calves [12]. Diagnosis of BVDV-associated abortion requires viral identification in conjunction with compatible histopathological findings [13]. In summary, the detection of these intracellular infectious agents and the presence of compatible lesions are essential to confirm their involvement as the cause of abortion [9]. However, histological inflammatory lesions indicative of infection may be present in fetal tissues even when it is not possible to detect a specific infectious agent [3,5,14]. Moreover, although the clinical and histopathological features of bovine abortion associated with these three intracellular agents have been described [3,5,9,10], there is limited information available regarding the immunological response in cases of natural abortion.

In the pathogenesis of abortion, the development of the fetal immune system plays a crucial role. From day 75 of gestation, the fetus can initiate its own immune response, which becomes fully developed by day 130 [15]. Despite this physiological development, the ontogeny of fetal immunity is generally stimulated by antigenic challenge [16]. Additionally, the maternal immune system must maintain a delicate balance between tolerating the conceptus (as an allograft) and responding to abortifacient agents. Therefore, the timing of infection may determine the clinical outcome, potentially resulting in persistent infection or fetal death [17]. Upon activation by pro-inflammatory cytokines (IL-1, TNF-α, and IL-6), local effector cells of the innate immune systems such as endothelial cells, epithelial cells, macrophages, and dendritic cells produce additional cytokines and chemokines that attract neutrophils and monocytes. This cascade subsequently recruits dendritic cells and NK, T, and B lymphocytes. In contrast, cytokines such as IL-4, IL-10, and IL-17 actively promote the resolution of inflammation [18]. Thus, immune responses can be classified by cytokine expression patterns as Th1 (IFN-γ, IL-2, IL-8, IL-12, TNF-α) or Th2 (IL-4, IL-5, IL-10) [18,19]. Moreover, various acute phase proteins (APPs) are produced in response to infection and inflammation, enhancing neutrophil and macrophage activity and promoting cytokine release [20,21]. Haptoglobin, synthesized in the liver, is considered one of the most specific APPs in cattle. Its concentration increases during inflammatory events and is being investigated as a potential biomarker for various diseases [22,23]. Fetal body fluids are easily accessible samples and may be useful for studying physiological and pathological changes during gestation, particularly from the second trimester (>120 days), when the fetus is immunocompetent. The presence of specific antibodies in fetal fluids indicates fetal exposure and immune response to antigens, making them a preferred sample for abortion diagnosis [5,14,24]. However, to our knowledge, there are no reports evaluating haptoglobin levels in fetal fluids from naturally aborted fetuses.

The objective of this study is to describe the pathological findings, cytokine expression, and haptoglobin levels in mid- and late-gestation fetuses aborted due to natural infections with intracellular agents, and to compare them with fetuses presenting inflammatory lesions without an etiological diagnosis (probable infectious etiology) and with negative control fetuses from the same gestational stages. This assessment may offer an alternative diagnostic approach to better understand the physiological and pathological processes occurring during bovine gestation.

## 2. Materials and Methods

### 2.1. Experiment Design and Selection of Fetuses

Aborted bovine fetuses submitted during 2022–2023 to the Specialized Veterinary Diagnostic Service (SVDS) of INTA Balcarce, Argentina, were routinely processed to achieve an etiological diagnosis. Fetuses aborted due to intracellular agents (*Neospora caninum*, *Brucella abortus*, and Bovine Viral Diarrhea Virus) were selected and are hereafter referred to as the “intracellular agent abortion” group. Additionally, fetuses with inflammatory lesions compatible with an infectious cause of abortion but without a definitive diagnosis of any specific agent were included as the “probable infectious etiology” group. Finally, a control group consisted of bovine fetuses recovered from abattoirs, showing no inflammatory lesions and no detectable infectious agents. Fetuses were classified by gestational age into mid gestation (4 to 6 months) and late gestation (>7 months), according to the morphological guidelines described by Kirkbride [25]. Post-mortem examinations and sample collection were performed following the standard operating procedures of INTA Balcarce [5]. Gross pathological findings were recorded during necropsy. The criteria used to confirm the etiological cause of abortion included: (1) direct detection of intracellular infectious agents in fetal tissues or fluids; (2) presence of inflammatory changes consistent with infection; and (3) exclusion of other abortifacient agents [5].

Due to the limited number of confirmed cases of *B. abortus* and BVDV-abortions, pathogen-specific comparisons were not feasible. Therefore, these abortions were grouped with *N. caninum*-abortions under the category of intracellular pathogens to explore general trends in gestational age-dependent immune modulation. This strategy reflects the exploratory nature of the study and was necessary to ensure sufficient sample representation for statistical analysis using randomization-based methods suitable for small datasets.

### 2.2. Diagnosis

#### 2.2.1. *Neospora caninum*

For the confirmation of *N. caninum*-associated abortion, DNA was extracted from fetal brain tissue using a commercially available kit (High Pure PCR Template Preparation Kit, Roche, Mannheim, Germany), following the manufacturer’s instructions. DNA concentration was measured using the Epoch microvolume spectrophotometer system (BioTek, Winooski, VT, USA), and samples were diluted to a final concentration of 60 ng/µL. For each PCR reaction, 5 µL of DNA were added, corresponding to a total of 300 ng of DNA per reaction, following the protocol described by Sánchez-Sánchez et al. [26] and Hecker et al. [27]. For molecular detection, a nested PCR targeting the internal transcribed spacer 1 (ITS-1) region of *N. caninum* was performed using four oligonucleotides, as described by Buxton et al. [28]. Secondary amplification products were visualized by electrophoresis on 2% agarose gels stained with SYBR Safe. DNA equivalent to 100 tachyzoites was used as a positive PCR control. Negative controls were included in each set of DNA extractions and PCR reactions.

To clarify the diagnostic criteria and analytical sensitivity of the assay, a sample was considered positive when a clear amplification product of the expected size (279 bp) was observed. The sensitivity of the assay, as established in our laboratory: 10 tachyzoites, determined through tenfold serial dilutions of DNA extracted from 100,000 tachyzoites (NC-1 strain) counted using a Neubauer chamber. All PCR runs included appropriate positive and negative controls to ensure reliability and rule out contamination.

#### 2.2.2. *Brucella abortus*

For the isolation of *B. abortus*, fetal lung, abomasal contents, and placenta (when available) were collected [29,30] and cultured on Columbia Blood Agar (CBA; Oxoid Ltd., Wad Road, Basingstoke, UK) supplemented with 5% sterile defibrinated bovine blood, and on Skirrow (SK) agar containing antibiotics [30]. CBA and SK plates were incubated at 37 °C in a 10% CO_2_ atmosphere and examined daily for bacterial growth over a 7-day period. All isolates were identified using standard procedures [29]. Based on previous experience of our research group and published reports [1,31], this incubation time has proven sufficient for primary isolation from bovine fetal samples, which typically present high bacterial loads. We acknowledge that longer incubation periods (up to 14–21 days) may be required for samples with low bacterial burden, such as blood cultures, but under the conditions used, 7 days was adequate for reliable recovery.

*Brucella* was identified based on colony morphology, Gram staining, and conventional biochemical tests, including catalase, urease, and oxidase activities, following the protocols described by Alton et al. [29], a widely accepted reference for *Brucella* diagnostics.

All procedures involving the isolation and handling of *Brucella* strains were performed in a Class II biological safety cabinet, in compliance with biosafety level 2 standards. Appropriate personal protective equipment, including lab coats, gloves, and face protection, was used throughout to mitigate zoonotic risk and ensure laboratory safety.

The genomic DNA was extracted for PCR assay from pure cultures of *Brucella* spp. single colonies were suspended in 50 μL sterile Milli-Q H_2_O and incubated in a 100 °C heating block for 20 min. Lysates were centrifuged and the cell-free supernatant was transferred to a fresh tube and DNA extracts were stored at 4 °C. PCR was used for the detection of the IS6501 of *Brucella* spp., as described by Saunders et al. [32], with modifications. Briefly, the IS6501 was amplified using the primers ISP1 and ISP2 under the following conditions: one cycle of denaturation at 94 °C for 2 min, 35 cycles of denaturation at 94 °C for 30 s, annealing at 55 °C for 30 s and extension at 72 °C for 1 min; generating a 600 bp amplification product. The mixture for the amplification consisted of 10 μL of 5× reaction buffer (Promega, Madison, WI, USA), 2.5 mM MgCl_2_, 0.2 mM of each deoxynucleotide triphosphate (dNTP), 200 μM of each primer, 5 μL of DNA template, and 3 U of Taq polymerase (Promega, Madison, WI, USA) and water to a final volume of 50 μL. DNA of *B. abortus* strain 2308 was used as positive controls. PCR products were separated into 1.0% agarose gel and stained with SYBR Safe (Invitrogen, Waltham, MA, USA).

#### 2.2.3. Bovine Viral Diarrhea Virus

BVDV isolation was performed on spleen, brain, and lung tissue samples according to standard procedures [33,34,35]. Briefly, a 10% tissue homogenate was prepared and inoculated into Madin-Darby Bovine Kidney cells, which were cultured at 37 °C in a 5% CO_2_ atmosphere. MDBK cells were obtained from the American Type Culture Collection (ATCC) and provided by the Virology Laboratory of the SVDS, INTA Balcarce, Argentina. Virus isolation was carried out through four blind passages, each lasting 72 h, which is the standard interval used in our laboratory to allow sufficient viral replication. After the final passage, BVDV antigen detection was performed using a direct immunofluorescence assay with a FITC-conjugated porcine polyclonal antiserum specific to BVDV (VMRD Inc., Pullman, WA, USA).

#### 2.2.4. Other Pathogens

Following the same diagnostic methodologies used for BVDV detection, the presence of bovine alphaherpesvirus was ruled out. Additionally, the absence of neutralizing antibodies in fetal cavity fluids against bovine alphaherpesvirus was confirmed by viral neutralization, as described by Spetter et al. [13]. To exclude *Campylobacter* spp., Skirrow agar supplemented with antibiotics (ASK) was used under microaerophilic conditions (85% N_2_, 10% CO_2_, 5% O_2_) [30]. In our study, the ASK medium contained vancomycin (10 µg/mL), polymyxin B (1.25 IU/mL), cycloheximide (50 µg/mL), and trimethoprim (5 µg/mL), all purchased from Sigma-Aldrich (St. Louis, MO, USA). Furthermore, MacConkey (MC) agar under aerobic conditions (Oxoid Ltd., Wad Road, Basingstoke, UK) was used to identify members of the Enterobacteriaceae family. For the diagnosis of *Leptospira* spp., kidney samples were processed using PCR protocols [36]. To detect *Tritrichomonas foetus*, abomasal fluid samples were inoculated into liver broth medium, incubated at 37 °C, and examined daily under a light microscope at 20× magnification for seven consecutive days [37]. If pathological findings suggestive of fungal infection were observed, samples were cultured on Sabouraud dextrose agar following the criteria established by Morrell et al. [5].

### 2.3. Histopathology

Tissue samples from the lung, encephalon (including cerebrum, cerebellum, and medulla oblongata), mesenteric and retropharyngeal lymph nodes, heart, thymus, kidney, liver, skeletal muscle, tongue, abomasum, small and large intestines, spleen, adrenal glands, skin, thyroid, and placenta (when available) were fixed in 10% buffered formalin (pH 7.2) for 72 h. Samples were approximately 0.5–1 cm thick to ensure adequate fixation. Sections were cut at 4 µm and stained with hematoxylin and eosin (HE).

### 2.4. Haptoglobin Determination

Fetal cavity fluid samples were stored at −80 °C until analysis. Haptoglobin concentration was determined in a single assay using a commercial turbidity test (Turbitest AA Line, Wiener Lab, Rosario, Argentina). Measurements were performed using an Intelligent Clinical Chemistry Analyzer (InCCA, DICONEX S.A., Quilmes, Argentina). For calibration, a four-point curve was generated using the High-Level Protein Calibrator (Wiener Laboratories, Rosario, Argentina) and 0.9% *w*/*v* NaCl physiological solution. Quality control was conducted using the undiluted High-Level Protein Calibrator, following the manufacturer’s recommendations.

### 2.5. Cytokine Gene Expression

Fetal spleens from selected aborted fetuses were collected and classified into mid- and late gestation groups, then compared with negative control fetuses of the same gestational age. Transcriptional expressions of bovine IFN-γ, TNF-α, IL-4, IL-8, and IL-12 genes were assessed in spleen tissue by real-time RT-qPCR using specific primers (Table 1). Briefly, spleen samples were stored at −80 °C until analysis. Total RNA was extracted using Trizol reagent (Invitrogen, Life Technologies Corporation, Carlsbad, CA, USA) and treated with DNase I Amplification Grade (Invitrogen) for 30 min at 37 °C to eliminate contaminating genomic DNA (gDNA). Complementary DNA (cDNA) was synthesized using 1 μg of total RNA, random hexamers (12 ng/μL) (Promega), and Moloney murine leukemia virus reverse transcriptase (10 U/μL) (Promega). Two technical replicates were included for each sample and negative controls omitting RNA or reverse transcriptase were included. RNA purity was assessed spectrophotometrically by measuring the A260/280 ratio using an Epoch microplate spectrophotometer (BioTek), and only samples within the accepted range for high-quality RNA (with A260/280 ratio values of 1.6 to 2) were processed. Furthermore, RNA integrity was assessed using the Ct value of the reference gene as an internal quality control criterion; only samples with a Ct ≤ 30 were included in the analysis, ensuring suitability for reliable quantification of gene expression [38,39]. Expression of the housekeeping gene glyceraldehyde-3-phosphate dehydrogenase (GAPDH) was used as an internal control [40]. RT-qPCR reactions contained 800 nM of each primer, 1× qPCR Master Mix (FastStart Universal SYBR Green Master Rox, Roche), and 1 μL of cDNA in a final volume of 20 μL. Amplification and detection were performed on an Applied Biosystems 7500 system under the following conditions: 2 min at 50 °C, 10 min at 95 °C, followed by 40 cycles of 20 s at 95 °C and 60 s at 60 °C. A melting curve analysis confirmed the specificity of the amplified products. Negative controls for cDNA synthesis and qPCR were included in all runs. Amplification efficiency for each gene was determined using 10-fold serial dilutions of cDNA. Results were expressed as the mean fold change in mRNA levels relative to control fetuses.

### 2.6. Statistical Analysis

Statistical comparisons were performed between fetuses naturally aborted due to infections by intracellular agents and fetuses recovered from abattoirs without detected infections or inflammatory lesions (control group); between fetuses naturally aborted due to probable infectious etiology (PIE) and the control group; and between fetuses aborted due to intracellular agents and those aborted due to PIE (which served as the control group in this comparison). All comparisons were conducted between fetuses of the same gestational age, grouped into mid-gestation fetuses (MGF) or late-gestation fetuses (LGF), as previously described. One mid-gestation control fetus was excluded from the gene expression analysis due to a Ct value > 30 for the reference gene, indicating insufficient RNA integrity for reliable quantification. The relative expression analysis of the target genes was performed using the Relative Expression Software Tool (REST-2009, Qiagen Inc., Valencia, CA, USA) for evaluating group differences for significance with a pair-wise fixed reallocation randomization [46]. The RT-qPCR efficiency for each gene was determined by a linear regression model according to the equation: E = 10 [−1/slope]. For gene expression data, conventional parametric assumptions were not applied. REST uses a randomization-based approach that does not require normal distribution and is specifically designed for small sample sizes. Therefore, conventional outlier detection was not performed on these data. The Ct values obtained after the various RT-qPCRs, as well as the exact *p*-values and additional details of the REST analysis for each sample and gene, are provided in Appendix A.

For haptoglobin analysis, data was analyzed using R Studio statistical software, version R-4.4.0. A one-way analysis of variance (ANOVA) was performed to assess differences in haptoglobin levels between groups of same gestational age. Model assumptions were evaluated by testing for normality of residuals using the Shapiro–Wilk test and inspecting Q-Q plots and residual-vs-fitted value plots. The presence of influential outliers was assessed using the studentized residuals test implemented in the outlierTest function of the car package (R software). A single outlier was identified, and was excluded from the analysis (rstudent = −5.96; Bonferroni *p* < 0.001). To evaluate its potential impact on the results, the analyses were repeated both including and excluding this observation. Levene’s test and the Breusch-Pagan test were used to assess homoscedasticity. When significant differences were found, post hoc comparisons were conducted using pairwise comparisons via estimated marginal means (emmeans package). The associations between haptoglobin concentration and autolysis score, as well as between cytokine levels and autolysis score, were assessed using Spearman’s rank correlation test, given the ordinal nature of the autolysis variable. Data visualization was carried out using Graph Pad Prism software (Graph Pad 4.0.3, GraphPad Prism Inc., San Diego, CA, USA).

## 3. Results

### 3.1. Fetuses Aborted Due to Intracellular Agents and Probable Infectious Etiology

Etiological diagnosis was confirmed in 40 (47.06%) out of 85 bovine fetuses received at INTA Balcarce. Among these, 25 (29.41%) fetuses were associated with intracellular agents, and 13 cases that met the selection criteria were included in the “intracellular agent abortion” group. The remaining 45 (52.94%) fetuses had no confirmed etiological diagnosis; however, 27 of them (31.76%) showed inflammatory lesions compatible with an infectious agent. Five of these 27 cases met the selection criteria and were included in the “probable infectious etiology” group. Within the “intracellular agent abortion” group, 9 cases were due to *N. caninum*, 2 to *B. abortus*, and 2 to BVDV. Twelve and six fetuses came from beef and dairy herds, respectively. Among them, 9 were male and 9 were female. Most fetuses (*n* = 12) were in the third trimester of gestation, while the remaining 6 were in the second trimester. Unfortunately, no mid-gestation fetuses with probable infectious etiology met the selection criteria.

Appendix A summarizes the causes of abortion, gestational age of the aborted fetuses, production system, sex, and the most relevant macroscopic and microscopic lesions. Additionally, 10 fetuses in mid- and late gestation (*n* = 3 and *n* = 7, respectively) were selected as control groups. The main histopathological findings are shown in Figure 1.

### 3.2. Haptoglobin

Haptoglobin concentrations are shown in Table 2. No significant differences in haptoglobin concentrations were observed among aborted fetuses at late gestation (Intracellular agents vs. Control vs. PIE) (F(2,16) = 0.45, *p* = 0.64; Table 3). The assumptions of normality (Shapiro–Wilk: W = 0.953, *p* = 0.450) and homoscedasticity (Levene’s Test: *p* = 0.97) were met. A post hoc power analysis was conducted to estimate the probability of detecting a true effect if it exists in the population., yielding a statistical power of 0.11 (11%). In this case, the low power indicates that the test had a limited ability to identify significant differences, suggesting that the study may be underpowered for this specific comparison. With a power of approximately 11%, the probability of detecting a significant difference (if one exists) in your analysis of variance is very low. This is usually an indication that the sample size (*n* = 5 per group) is too small to detect a moderate effect (f = 0.25), or that the effect size used in the calculation (f = 0.25) is too small to be detected with the current samples.

However, aborted fetuses at mid-gestation due to intracellular agents had lower haptoglobin concentrations compared to control fetuses (F(1,6) = 8.06, *p* = 0.03; see Figure 2). The assumption of normality of the residuals was met (Shapiro–Wilk: W = 0.912, *p* = 0.368), as was the assumption of homogeneity of variance (Levene’s Test: *p* = 0.78). Tukey’s post hoc comparisons indicated that values were higher in the Control group than in the Intracellular group (mean difference = 9.87 mg/dL, 95% CI [1.36, 18.38]).

No significant association was found between haptoglobin concentration and autolysis score (Spearman’s rho = −0.19, *p* = 0.33).

### 3.3. Cytokine Gene Expression in Fetal Spleen

No significant differences in TNFα, IL-4, and IL-12 gene expression were detected in the spleens of aborted fetuses at mid- and late gestation (intracellular agents vs. control) (*p* > 0.05). However, mid-gestation fetuses aborted due to intracellular agents showed a significant upregulation of IFN-γ (10.66-fold change) and IL-8 (9.43-fold change) compared to control fetuses (*p* < 0.05). In contrast, late-gestation fetuses aborted due to intracellular agents showed a significant downregulation of IFN-γ (0.05-fold change) and IL-8 (0.06-fold change) compared to controls (*p* < 0.05) (Figure 3, Appendix A). No significant differences in TNFα, IL-4, IL-8, and IL-12 gene expression were observed in the spleens of fetuses aborted due to probable infectious causes (*p* > 0.05). However, late-gestation bovine fetuses aborted due to probable infectious causes showed a significant downregulation of IFN-γ (0.017-fold change) compared to controls (*p* < 0.05) (Figure 4, Appendix A). Finally, late-gestation fetuses aborted due to intracellular agents showed a significant upregulation of IL-4 (12.34-fold change, Th2) and a downregulation of IL-8 (0.07-fold change) compared to fetuses aborted due to probable infectious causes (*p* < 0.05) (Figure 5, Appendix A). The results, represented in Figure 3, Figure 4 and Figure 5, show the mean fold change in gene transcription levels in mid- and late-gestation fetuses.

No significant association was found between cytokine concentration and autolysis score (Spearman’s rho = −0.19, *p* = 0.32).

## 4. Discussion

The present study explores the immune response in fetuses spontaneously aborted during mid- and late gestation due to infections caused by intracellular agents or of probable infectious etiology. However, no mid-gestation fetuses met the inclusion criteria for the probable infectious etiology group, limiting direct comparisons across all gestational stages. Consequently, interpretations involving this group are restricted to late gestation. This limitation reflects the strict diagnostic and quality criteria applied and underscores the preliminary exploratory nature of the findings. In this study, priority was given to the evaluation of cytokines with well-established roles in the immune response to intracellular agents (such as IFN-γ, IL-4, and IL-8), given their relevance in the differentiation of Th1 and Th2 profiles and their involvement in fetal immunity. Although the sample size was limited due to stringent selection criteria, this work represents an initial step toward the immunological characterization of the fetus in cases of bovine abortion. Future studies with greater sample availability and broader resources could expand the cytokine panel to include IL-1, IL-6, and IL-10. While TNF-α and IL-12 were included in the cytokine expression analysis, no significant differences were observed among the study groups. This lack of modulation may reflect the temporal dynamics of cytokine expression, with these mediators potentially peaking at stages of infection not captured at the time of sampling. Alternatively, TNF-α and IL-12 may act in other fetal tissues or compartments beyond the spleen, or play more prominent roles in maternal-fetal immune interactions. Future investigations should consider longitudinal sampling and multi-tissue analysis to better elucidate the kinetics and localization of these cytokines during fetal infection. This approach is supported by previous studies emphasizing the importance of tissue-specific immune responses in fetal and maternal compartments [18,47]. Given the exploratory nature of this study and the limited sample size, cytokine expression was assessed through relative quantification methods. In future research, the use of ELISA assays could provide a more objective and quantitative measurement of cytokine concentrations, thereby enhancing the accuracy and reproducibility of the findings. Thus, the present study should be considered an initial approximation to fetal immune profiling in bovine abortions. Additionally, control fetuses were collected opportunistically from abattoirs and selected based on strict criteria: absence of inflammatory lesions, no detectable infectious agents, and gestational age matching. While this strategy aimed to minimize biological variability, we acknowledge that it may not fully account for potential confounders such as breed, health status, or management conditions. Future studies should incorporate more rigorous control selection protocols, including matched cohorts and formal sensitivity analyses to better address selection bias.

Spontaneously aborted fetuses may exhibit varying degrees of autolysis, characterized by distinct gross and histological changes [5,6,7,8,9,10,11,12,13,14], as fetal death often occurs several days or even weeks prior to expulsion, potentially interfering with diagnostic assessment. However, the fetuses selected for this study exhibited only minimal autolytic changes upon postmortem examination and histopathological evaluation. Additionally, GAPDH gene expression was used as a control to confirm the suitability of the selected fetuses for molecular analysis [40]. Importantly, no significant correlation was found between autolysis score and either cytokine Ct values or haptoglobin concentration, suggesting that sample quality did not confound the molecular or biochemical findings. Nevertheless, autolysis, although minimal in selected cases, may still affect RNA integrity and protein levels, potentially influencing the interpretation of molecular and biochemical findings. Pathological findings in all selected fetuses aborted due to intracellular agents were consistent with those previously described by Anderson [9], Campero et al. [3], and Morrell et al. [5]. Moreover, mid-gestation abortions mainly associated with *N. caninum* infections exhibited more severe inflammatory lesions compared to late-gestation *N. caninum* abortions, as previously reported by Dorsch et al. [48]. Additionally, mid- and late-gestation fetuses aborted due to intracellular agents showed significant upregulation of IFN-γ and downregulation of IL-8. These results may be associated with the onset of acute infection by intracellular agents during mid-gestation, characterized by a predominant Th1-type immune response. In this context, it is important to highlight that pregnancy hormones induce changes in immune cell populations and promote immunological tolerance of the fetus, shifting the immune environment toward a Th2 profile during mid-pregnancy, along with immune suppression [18], which may increase fetal susceptibility to infectious processes. As mentioned above, late-gestation fetuses aborted due to intracellular agents exhibited a distinct immunological profile compared to mid-gestation fetuses. In these cases, infection may have occurred several weeks prior to fetal death, allowing time for the development of a fetal immune response. This supports the notion that we are likely observing only a partial picture, as the timing of infection and abortion do not necessarily coincide. Furthermore, although late-gestation fetuses of probable infectious etiology presented lesions similar to those observed in fetuses aborted due to intracellular agents, their cytokine expression patterns differed significantly, suggesting the possible involvement of other pathogens in the former group. Specifically, significant upregulation of IL-4 and downregulation of IL-8 were observed in fetuses aborted due to intracellular agents compared to those of probable infectious etiology. These findings suggest that fetuses of probable infectious etiology may exhibit a stronger Th1-type response than those aborted due to confirmed intracellular infections.

There is strong evidence supporting the induction of Th1 cytokine mRNA and its role in regulating cellular immune responses against intracellular infections [49]. The expression and production of cytokines have been evaluated in experimental *N. caninum* infections to better understand the mechanisms underlying abortion, revealing upregulated expression and production of IFN-γ [50,51]. In this context, it has been proposed that dams with high titers of IgG2 antibodies and concurrent IFN-γ production may be protected against abortions caused by *N. caninum* [52]. Additionally, both macrophages and dendritic cells exposed to viable *N. caninum* tachyzoites have been shown to elicit a Th1-type response characterized by the secretion of IFN-γ and IL-12 [53].

Similarly, increased expression of IFN-γ, IL-1β, and IL-6 has been observed in bovine placental and blood samples from dams infected with *B. abortus* [54,55]. In murine models, *Brucella* infections elicit a Th1-type cellular immune response that promotes bacterial clearance. The development of this response is regulated by key cytokines such as TNF-α, IFN-γ, and IL-12, produced during the early stages of infection [56,57].

Furthermore, partial inhibition of type I interferon (IFN-I) production has been observed in vitro in bovine cells infected with BVDV, facilitating viral replication [58]. This inhibition may contribute to the establishment of persistent BVDV infection, although it contrasts with previous studies reporting a significant increase in IFN-I production in the blood of heifers and fetuses infected with BVDV [59,60]. However, it has also been shown that fetuses infected with BVDV can mount an adaptive immune response characterized by increased IFN-γ expression, which partially controls fetal viremia [61].

Acute phase proteins (APPs) have also been detected in bovine fetuses during infections and reproductive losses [14]. Although serum amyloid A (SAA) is an important biomarker in cattle [62], haptoglobin was selected due to its higher specificity in this species and the availability of validated methods in our laboratory. Haptoglobin is an early biomarker of acute inflammation, with potential diagnostic and prognostic value, as well as possible utility in monitoring treatment response [23]. Additionally, haptoglobin inhibits microbial growth during infection by sequestering iron within hemoglobin [22]. In this context, significantly increased haptoglobin levels have been observed in cows with abortions and dystocia compared to those with normal pregnancies [63]. Similarly, Çenesiz et al. [64] reported a significant increase in serum haptoglobin in cattle infected with *B. abortus*. Conversely, perinatal death in calves associated with uterine infection did not show significant differences in plasma haptoglobin concentrations compared to live calves [65]. It is also important to note that haptoglobin is present in the reproductive tissues and fluids of healthy animals and has been associated with reduced oxidative stress and benefits for embryonic development [66]. Some studies have reported a significant increase in other APPs, such as SAA and IL-6, in the blood and plasma of aborted bovine fetuses and perinatal deaths due to infectious causes, suggesting their potential as biomarkers for reproductive losses [62,67]. In our study, significant differences were observed only in mid-gestation fetuses, with higher haptoglobin values in control animals. Beyond these mid-gestation differences, haptoglobin concentrations remained low, suggesting limited biological significance and raising questions about the suitability of fetal body cavity fluids for haptoglobin analysis as a diagnostic tool. Although the Turbitest AA Line assay (Wiener Lab) is routinely used in our diagnostic laboratory and has demonstrated reliable performance in bovine serum and body fluids, its application to fetal cavity fluids, has not been formally validated. Therefore, the diagnostic interpretation of haptoglobin levels in this matrix should be considered provisional, pending analytical validation including limit of detection, limit of quantification, linearity, recovery, and assay variability. Future studies should consider spike-and-recovery experiments to confirm assay sensitivity and accuracy in this matrix, as recommended for diagnostic biomarkers in non-standard biological samples [68]. It is important to note that the elevated haptoglobin levels observed in mid-gestation control fetuses may reflect developmental regulation rather than immunological responses to infection. This introduces a potential confounding factor when interpreting the data. Given that control fetuses were opportunistically collected and matched by gestational age, but not by other variables such as breed or maternal condition, we acknowledge the limitations in control group design. Future studies should incorporate more rigorously selected gestational age-matched controls to better assess the ontogeny of acute phase proteins such as haptoglobin.

## 5. Conclusions

This study provides novel insights into the fetal immune response during bovine abortions caused by intracellular pathogens and probable infectious etiologies, highlighting gestational age as a key modulator of cytokine expression. The differential expression of IFN-γ and IL-8 suggests a dynamic shift in immune polarization, with a Th1-type response predominating in mid-gestation and a Th2-type profile emerging in late gestation. Although haptoglobin levels showed some variation, their diagnostic utility in fetal body fluids appears limited. The diagnostic utility of haptoglobin in fetal fluids remains provisional due to the lack of formal analytical validation of the assay for this specific matrix. Future studies should address this limitation through comprehensive validation protocols. Importantly, the absence of significant modulation in TNF-α and IL-12 may reflect temporal or tissue-specific expression patterns not captured in this study. Given the exploratory nature and limited sample size, these findings should be interpreted as preliminary. Future studies incorporating larger cohorts, longitudinal sampling, multi-tissue analysis, and quantitative protein-level assays such as ELISA are warranted to validate and expand upon these observations. Ultimately, a deeper understanding of fetal immunobiology may enhance diagnostic strategies and inform preventive approaches for reproductive losses in cattle.

## Figures and Tables

**Figure 1 animals-15-02878-f001:**
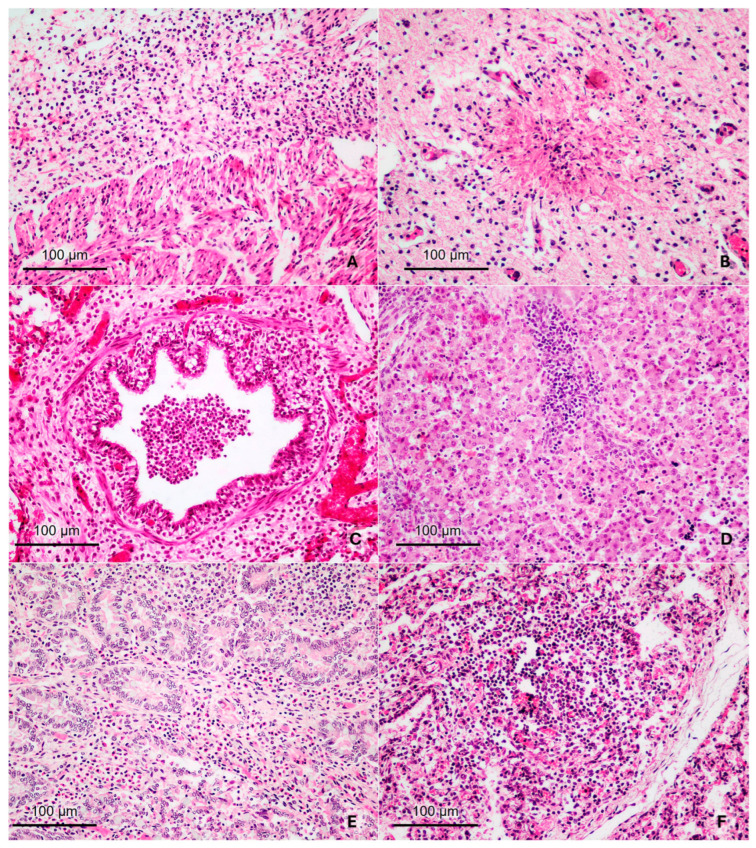
Histopathological findings in aborted bovine fetuses at mid- and late gestation due to intracellular infectious agents and probable infectious abortion (all images are at 200× magnification). (**A**) Bovine fetus aborted due to *N. caninum* infection, at 4 months of gestation (Fetus # 4). Heart: Severe lympho-histiocytic epicarditis. (**B**) Bovine fetus aborted due to *N. caninum* infection, full-term gestation (Fetus # 9). Brainstem: necrotizing encephalitis characterized by focal areas of necrosis and neuropil degeneration, surrounded by mononuclear infiltrate. (**C**) Bovine fetus aborted due to *B. abortus* infection, full-term gestation (Fetus # 11). Lung: abundant macrophages in bronchioles and histiocytic peribronchiolar infiltration. (**D**) Bovine fetus aborted due to BVDV infection, 5 months of gestation (Fetus # 12). Liver: portal hepatitis characterized by the presence of moderate lympho-histiocytic infiltration. (**E**) Bovine fetus of probable infectious etiology, full-term gestation (Fetus # 17). Small intestine: diffuse interstitial histiocytic infiltration in the mucosa. (**F**) Bovine fetus of probable infectious etiology, full-term gestation (Fetus # 18). Lung: Severe multifocal lympho-histiocytic infiltration in the alveolar septum.

**Figure 2 animals-15-02878-f002:**
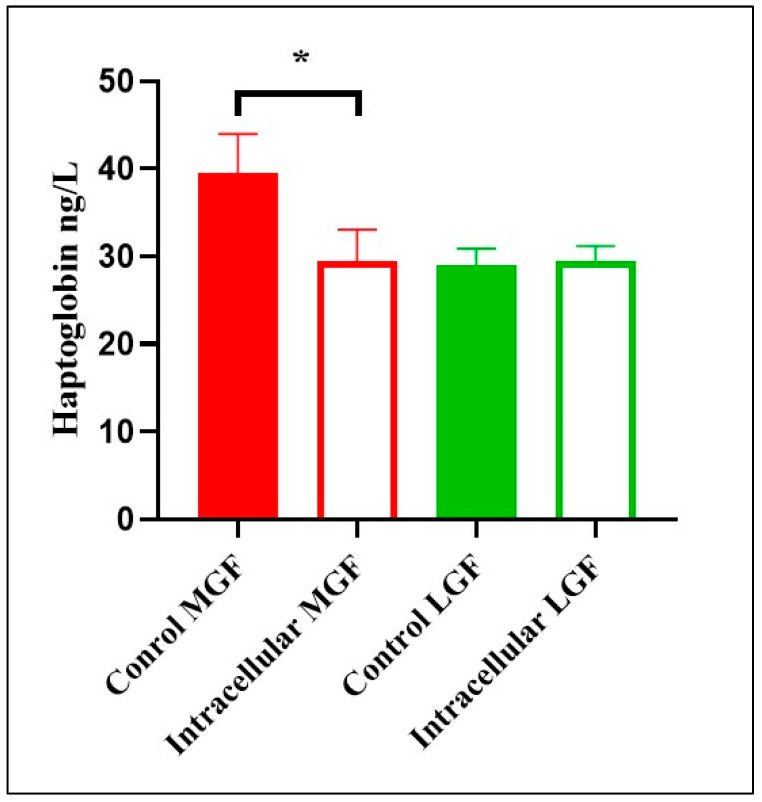
Haptoglobin concentrations in body cavity fluids of bovine fetuses naturally aborted due to intracellular agents and control fetuses at mid (MGF) and late gestation (LGF). *: Statistically significant differences (*p* < 0.05).

**Figure 3 animals-15-02878-f003:**
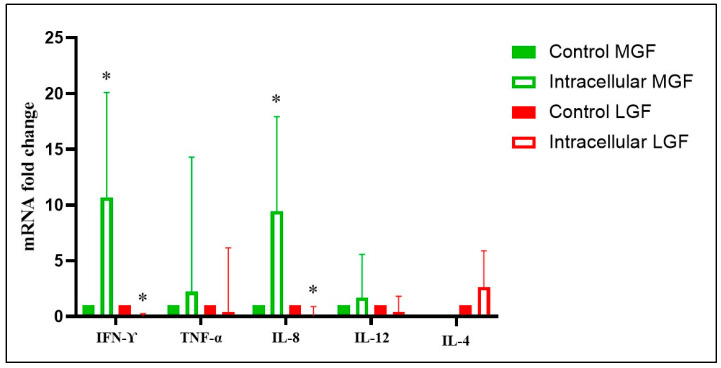
Relative expression of IFN-γ, TNFα, IL-8, IL-12, and IL-4 in the spleens of bovine fetuses naturally aborted due to intracellular agents at mid (MGF) and late gestation (LGF). The results represent the mean variation in transcript levels of IFN-γ, TNFα, IL-4, IL-8, and IL-12 in bovine fetuses aborted by intracellular agents at mid-gestation (MGF) and late gestation (LGF), compared to fetuses recovered from abattoirs without detected infections or inflammatory lesions (control group) at mid-gestation and late gestation (MGF control and LGF control, respectively, value 1). For each sample, cDNA was analyzed in two technical replicates, and the mean Ct value was used for statistical analysis. *: Statistically significant differences (*p* < 0.05).

**Figure 4 animals-15-02878-f004:**
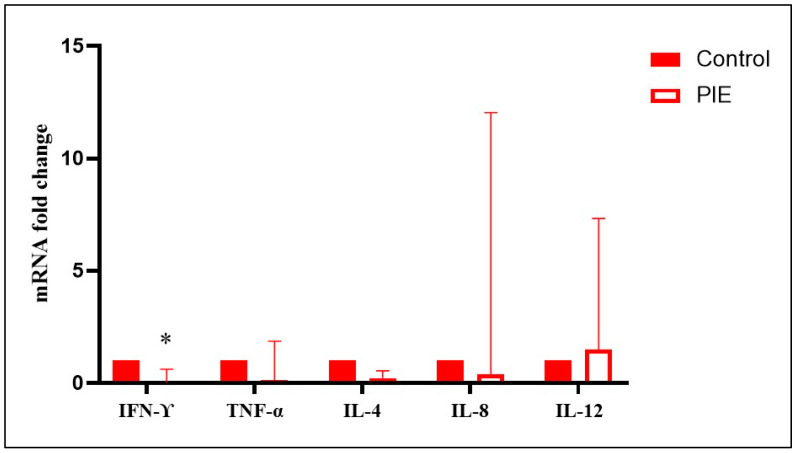
Relative expression of IFN-γ, TNFα, IL-4, IL-8, and IL-12 in the spleens of bovine fetuses naturally aborted due to probable infectious etiology (PIE). The results represent the mean change in transcript levels of IFN-γ, TNFα, IL-4, IL-8, and IL-12 in PIE fetuses at late gestation (LGF) relative to fetuses recovered from abattoirs without detected infections or inflammatory lesions (LGF control, value 1). For each sample, cDNA was analyzed in two technical replicates, and the mean Ct value was used for statistical analysis. *: Statistically significant differences (*p* < 0.05).

**Figure 5 animals-15-02878-f005:**
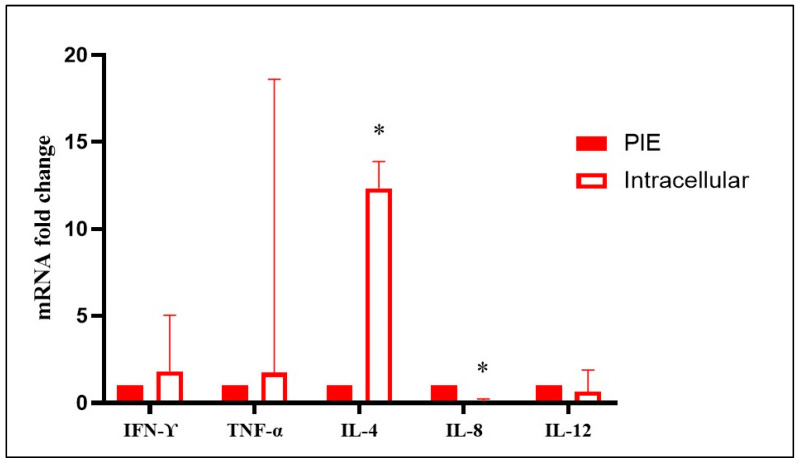
Relative expression of IFN-γ, TNFα, IL-4, IL-8, and IL-12 in the spleens of bovine fetuses aborted due to intracellular agents. The results represent the mean variation in transcript levels of IFN-γ, TNFα, IL-4, IL-8, and IL-12 in bovine fetuses aborted by intracellular agents compared with fetuses aborted naturally due to probable infectious etiology (PIE, value 1). For each sample, cDNA was analyzed in two technical replicates, and the mean Ct value was used for statistical analysis. *: Statistically significant differences (*p* < 0.05).

**Table 1 animals-15-02878-t001:** Primers used for evaluation of cytokine expression in tissues from bovine abortions and control fetuses.

mRNA	Primer Sense	Size (pb)	5′-3′ Sequence	Reference	NM Reference
GAPDH	F	112	TTCTGGCAAAGTGGACATCGT	[40]	NM_001034034.2
	R		CTTGACTGTGCCGTTGAACTTG		
IL-4	F	83	CATGCATGGAGCTGCCTGTA	[41]	NM_173921.2
	R		AATTCCAACCCTGCAGAAGGT		
IFN-γ	F	110	GATTCAAATTCCGGTGGATG	[42]	NM_174086.1
	R		TTCTCTTCCGCTTTCTGAGG		
TNFα	F	176	AGCCTCAAGTAACAAGCC	[43]	NM_173966.3
	R		TGAAGAGGACCTGTGAGT		
IL-8	F	60	GTTGCTCTCTTGGCAGCTTT	[44]	NM_173925.2
	R		GGTGGAAAGGTGTGGAATGT		
IL-12	F	157	AGTACACAGTGGAGTGTCAG	[45]	NM_174356.1
	R		TTCTTGGGTGGGTCTGGTTT		

**Table 2 animals-15-02878-t002:** Haptoglobin concentrations in fetal body cavity fluids during mid and late gestation.

Group.	Gestation Stage	Infectious Agent	Number	Hp (ng/L)
Intracellular agent	Mid-gestation	*N. caninum*	1	30.37
		*N. caninum*	2	27.36
		*N. caninum*	3	3.83
		*N. caninum*	4	36.13
		*B. abortus*	5	28.53
		BVDV	6	24.36
	Late-gestation	*N. caninum*	7	29.14
		*N. caninum*	8	28.53
		*N. caninum*	9	30.84
		*N. caninum*	10	31.64
		*N. caninum*	11	26.69
		*B. abortus*	12	30.73
		BVDV	13	28.98
Probable infectious abortion	Late-gestation	-	14	27.74
		-	15	31.11
		-	16	29.19
		-	17	27.27
		-	18	27.35
Negative control fetuses	Mid-gestation	-	19	44.95
		-	20	38.63
		-	21	34.09
	Late-gestation	-	22	30.68
		-	23	29.03
		-	24	29.93
		-	25	29.31
		-	26	29.96
		-	27	29.01
		-	28	24.73

Hp: Haptoglobin.

**Table 3 animals-15-02878-t003:** Haptoglobin concentrations in body cavity fluids of bovine fetuses naturally aborted due to intracellular agents, probable infectious etiology and control fetuses at late gestation (LGF).

	Haptoglobin Concentration (LGF)			SEM	*p*-Value
	Control	Intracellular	PIE		Group
Group	28.9	29.5	28.5	0.714	0.64
CI 95%	(27.5–30.4)	(28.1–30.9)	(26.8–30.2)		

## Data Availability

Data will be available under request.

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
