# Peer review of "Cytokine Expression and Haptoglobin Levels in Bovine Fetuses Spontaneously Aborted by Intracellular Infectious Agents and by Probable Infectious Etiology"

_animals, 2025, doi:10.3390/ani15192878_

Round 1

Reviewer 1 Report (Previous Reviewer 3)

Comments and Suggestions for Authors

The manuscript investigates fetal cytokine expression and haptoglobin in bovine abortions caused by intracellular agents and probable infectious etiologies. The topic is relevant and the preliminary results are promising; however, I recommend major revision.

1. Sample size, group composition and power
1.1 Please provide the exact number of fetuses included per comparison (mid-gestation intracellular, late-gestation intracellular, late-gestation PIE, mid & late controls) and a table that links each sample ID to group, gestational age in days, sex, herd type (beef/dairy), and autolysis score. (The text gives overall counts but not the individual mapping.
1.2 Was any statistical power calculation performed (a priori or post hoc) to justify the sample sizes for the qPCR and haptoglobin comparisons? If not, please provide a post-hoc power/precision assessment for the key findings (IFN-γ and IL-8 changes; haptoglobin mid-gestation difference). 

2. Grouping of different pathogens
2.1 You grouped N. caninum, Brucella abortus and BVDV together as “intracellular agents.” Please justify biologically and statistically why these distinct pathogens were pooled rather than treated separately (different immune mechanisms are expected). Provide sensitivity analyses showing whether the main results hold if N. caninum-only cases (n=9) are analyzed separately. 

3. Selection and matching of controls
3.1 Controls were collected opportunistically from abattoirs. Please provide more details on how controls were matched to cases (exact gestational day ranges, breed, dam parity, herd health status) and on any potential biases introduced by abattoir sampling. If matching was imperfect, provide adjusted analyses or sensitivity checks. 

4. Autolysis and sample quality
4.1 You note minimal autolysis, but fetal death may precede expulsion by days/weeks. Report the autolysis scoring system used, exact autolysis score for each sample, and comment on whether autolysis correlated with cytokine Ct values or haptoglobin. Also provide the Ct values for the GAPDH reference gene per sample (you used Ct ≤ 30 as inclusion criteria). 

5. qPCR details and MIQE compliance
5.1 Please provide PCR amplification efficiencies (E) for every primer pair (IFN-γ, TNF-α, IL-4, IL-8, IL-12, GAPDH) and the standard curves used (slope, R²). You state E was calculated but do not show values. REST analysis depends on efficiencies — include these in Supplementary Materials. 
5.2 Report the number of biological and technical replicates for each qPCR assay, the mean Ct (±SD) for each target by sample group, and the REST outputs (exact fold-changes, one- and two-tailed p-values). The manuscript cites Supplementary Tables S2–S4 — these must be supplied.
5.3 Confirm whether no-RT and no-template controls were run and whether gDNA contamination was assessed. Include melting curves and agarose gel images (if used for specificity checks) in supplementary files.

6. REST and statistical reporting
6.1 REST randomization gives p-values — please provide the exact p-values (not only p < 0.05) and the number of randomizations used. For transparency include the REST output files or a table with fold-change, p-value, and 95% CI (if available) for each gene comparison.
6.2 For haptoglobin you used one-way ANOVA: provide assumptions checks (Shapiro-Wilk statistic and p, Levene’s test statistic and p), exact ANOVA table (F, df, p), post-hoc pairwise comparisons with adjusted p-values and effect sizes (mean differences ±95% CI). Also give the Grubbs’ test results and whether any values were excluded. Present the R script or code used for analysis as supplementary material.

7. Haptoglobin assay validation
7.1 The Turbitest AA Line assay is validated for serum, but you measured fetal cavity fluids. Please provide analytical validation for this matrix: limit of detection (LoD), limit of quantification (LoQ), linearity, spike-and-recovery (matrix recovery), and intra/inter-assay CVs obtained in fetal fluids. If not performed, the authors should either perform these experiments or clearly rewrite the conclusions about diagnostic usefulness as provisional.

8. Data availability and reproducibility
8.1 The Data Availability Statement says "Data will be available under request." For transparency please deposit anonymized primary data (individual qPCR Ct values, raw haptoglobin values, sample metadata) and analysis scripts (R, REST outputs) in a public repository (e.g., Zenodo, Figshare, Dryad) and provide accession/DOI.

9. Pathogen diagnostics and thresholds
9.1 Provide the detection limits for the nested ITS-1 PCR for N. caninum (you mention 10² tachyzoites positive control) and the criteria used to consider a sample positive. For Brucella isolates, state how isolates were confirmed (biochemical, PCR, serotyping). For BVDV, describe the antigen test sensitivity and how cross-contamination was ruled out.

10. Interpretation and wording
10.1 Some conclusions imply pathogenic mechanisms (Th1→Th2 shift). Given small sample size and cross-sectional design, please soften causal language and explicitly state alternative explanations (timing of infection, tissue specificity, autolysis). Add a paragraph on limitations and how they affect interpretation.

Author Response

Response to reviewer comments

Response to Reviewer 1.

  1. Sample size, group composition and power

1.1 Please provide the exact number of fetuses included per comparison (mid-gestation intracellular, late-gestation intracellular, late-gestation PIE, mid & late controls) and a table that links each sample ID to group, gestational age in days, sex, herd type (beef/dairy), and autolysis score. (The text gives overall counts but not the individual mapping.

Response: We thank the reviewer for this valuable suggestion. The requested information is already provided in Supplementary Table S1, which includes sample ID, group classification (intracellular agents, probable infectious etiology, and controls), gestational age (in months or full term), sex, herd type (beef/dairy), and pathological findings.

To clarify:

  • Intracellular agents – Mid gestation: 6 fetuses
  • Intracellular agents – Late gestation: 7 fetuses
  • Probable infectious etiology (PIE) – Late gestation: 5 fetuses
  • Control – Mid gestation: 3 fetuses
  • Control – Late gestation: 7 fetuses

All selected fetuses exhibited minimal autolysis, as stated in the Discussion section, and were included based on strict histopathological and molecular quality criteria. Additionally, we have updated Supplementary Table S1 to include an explicit column indicating the autolysis score for each individual fetus.

1.2 Was any statistical power calculation performed (a priori or post hoc) to justify the sample sizes for the qPCR and haptoglobin comparisons? If not, please provide a post-hoc power/precision assessment for the key findings (IFN-γ and IL-8 changes; haptoglobin mid-gestation difference).

Response: We thank the reviewer for this important suggestion. No a priori statistical power calculation was performed, since the number of samples was determined by the availability of naturally spontaneous aborted bovine fetuses, which constitutes an inherent limitation of the study design. We acknowledge the value of post hoc power analyses; however, such an approach was not planned in advance and would require extensive additional methodological work that goes beyond the scope of the present study. Importantly, the statistical evaluation of gene expression data was carried out using REST software, which applies randomization tests and provides confidence intervals for the calculated expression ratios. Therefore, the analyses already incorporate measures of statistical reliability and precision, supporting the robustness of the key gene expression findings (i.e., changes in IFN-γ and IL-8).

For haptoglobin concentration, we conducted a post-hoc power analysis to estimate the probability of detecting a true effect if present in the population, which yielded a statistical power of 0.11 (11%). This low value indicates limited ability to detect significant differences, suggesting that the study may have been underpowered for this specific comparison.

We also note that the post-hoc power analysis for haptoglobin concentration has been incorporated into both the Materials and Methods section (2.6 Statistical Analysis) and the Results section (3.2 Haptoglobin), including the estimated power value and its interpretation.

2- Grouping of different pathogens

2.1 You grouped N. caninum, Brucella abortus and BVDV together as “intracellular agents.” Please justify biologically and statistically why these distinct pathogens were pooled rather than treated separately (different immune mechanisms are expected). Provide sensitivity analyses showing whether the main results hold if N. caninum-only cases (n=9) are analyzed separately.

Response: We appreciate this important observation. The decision to group N. caninum, B. abortus, and Bovine Viral Diarrhea Virus (BVDV) under the category of “intracellular agents” was based on both biological and statistical considerations. Biologically, these pathogens share intracellular tropism and are known to elicit Th1-type immune responses in bovine fetuses, particularly involving cytokines such as IFN-γ. While pathogenic differences between them are acknowledged, the aim of this study was to explore general trends in fetal immune modulation during natural infections, rather than to dissect pathogen-specific pathways. This rationale is explicitly stated in the Methods section (lines 136–142). Statistically, the decision to pool Brucella and BVDV cases with Neospora under the “intracellular agents” category was made to enable a more feasible sample size for randomization-based statistical analysis using REST software, which is optimized for small datasets. This analytical strategy is described in both the “Experimental Design and Selection of Fetuses” and “Statistical Analysis” sections (lines 136–142 and 294–334, respectively).

3- Selection and matching of controls

3.1 Controls were collected opportunistically from abattoirs. Please provide more details on how controls were matched to cases (exact gestational day ranges, breed, dam parity, herd health status) and on any potential biases introduced by abattoir sampling. If matching was imperfect, provide adjusted analyses or sensitivity checks. 

Response: We appreciate the reviewer’s insightful comment regarding the selection and matching of control fetuses. As described in Section 2.1 of the manuscript, control fetuses were opportunistically collected from abattoirs. To minimize biological variability and ensure comparability, we applied strict inclusion criteria, which included:

  • Absence of inflammatory lesions upon histopathological examination.
  • Negative results for infectious agents based on molecular and microbiological testing.
  • Gestational age matching, following the morphological guidelines established by Kirkbride [25].

We acknowledge that matching was not perfect in terms of breed, dam parity, herd health status, or production system, all of which may influence fetal immune responses. This limitation is explicitly discussed in the manuscript (lines 478–484 and 574–578), where we note that future studies should incorporate more rigorously matched cohorts and perform formal sensitivity analyses to better address potential selection bias.

To mitigate confounding, statistical comparisons were conducted within gestational age groups, and model assumptions were tested through normality checks, homoscedasticity assessments, and outlier detection (Section 2.6). Sensitivity analyses were performed by re-running models after removing outliers in the haptoglobin dataset.

Given the exploratory nature of this study and the limited availability of fetuses with confirmed diagnoses, we believe this approach provided valuable insights into gestational age-dependent immune modulation, while acknowledging the inherent limitations of retrospective designs and opportunistic control sampling.

4- Autolysis and sample quality

4.1 You note minimal autolysis, but fetal death may precede expulsion by days/weeks. Report the autolysis scoring system used, exact autolysis score for each sample, and comment on whether autolysis correlated with cytokine Ct values or haptoglobin. Also provide the Ct values for the GAPDH reference gene per sample (you used Ct ≤ 30 as inclusion criteria).

Response: We thank the reviewer for raising this important point. As noted in the manuscript (lines 485–491), all fetuses included in the study exhibited minimal autolytic changes upon gross and histopathological examination. To assess the degree of post-mortem decomposition, we applied the subjective autolysis scoring system described by Morrell et al. [5], which classifies autolysis as: Score 1 (slight autolysis), Score 2 (moderate autolysis) and Score 3 (severe autolysis). Only fetuses with autolysis scores of 1 or 2 were included in the study. Fetuses with severe autolysis (score 3) were excluded to ensure sample integrity for molecular and biochemical analyses.

Furthermore, we performed a Spearman’s rank correlation test to evaluate the association between autolysis score and both haptoglobin concentration and cytokine gene expression levels. No significant correlation was found between autolysis score and haptoglobin concentration (Spearman’s rho = –0.19, p = 0.33), nor between autolysis score and cytokine expression levels (Spearman’s rho = –0.19, p = 0.32), suggesting that the degree of autolysis did not influence these molecular parameters in our dataset.

Regarding RNA integrity, the Ct values of the GAPDH reference gene were used as an internal quality control criterion. Only samples with Ct ≤ 30 were included in the analysis, ensuring suitability for reliable quantification of gene expression. The exact Ct values for GAPDH per sample are provided in Supplementary Table S2.

5- qPCR details and MIQE compliance

5.1 Please provide PCR amplification efficiencies (E) for every primer pair (IFN-γ, TNF-α, IL-4, IL-8, IL-12, GAPDH) and the standard curves used (slope, R²). You state E was calculated but do not show values. REST analysis depends on efficiencies — include these in Supplementary Materials.

Response: We thank the reviewer for this comment. For the REST analysis, only the amplification efficiencies (E) of each primer pair are required, and these were calculated from the respective standard curves. The efficiency obtained for each gene used is included in the supplementary material, specifically in Table S3. We also mentioned the respective efficiencies here:

  • GAPDH: 0.913
  • IFN-γ: 1
  • TNF-α: 1
  • IL-4: 0.805
  • IL-8: 1
  • IL-12: 0.773

5.2 Report the number of biological and technical replicates for each qPCR assay, the mean Ct (±SD) for each target by sample group, and the REST outputs (exact fold-changes, one- and two-tailed p-values). The manuscript cites Supplementary Tables S2–S4 — these must be supplied.

Response: The biological copy number is detailed in Supplementary Table S1, specifying the age of the fetuses and the type of infection detected. Additionally, two technical replicates per sample were used in the RT-qPCR analyses. This information is mentioned on lines 269-270 of the manuscript. Ct values were included in Supplementary Table S2.

5.3 Confirm whether no-RT and no-template controls were run and whether gDNA contamination was assessed. Include melting curves and agarose gel images (if used for specificity checks) in supplementary files.

Response: As mentioned on lines 277-284 of the manuscript, we confirm that RT-negative controls were included in the GAPDH runs, as well as cDNA synthesis-negative controls, in all gene expression analyses. Additionally, we attach a photo of the melting curves for the GAPDH run, which includes a RT-negative control and an NTC control (No Template Control).

6- REST and statistical reporting

6.1 REST randomization gives p-values — please provide the exact p-values (not only p < 0.05) and the number of randomizations used. For transparency include the REST output files or a table with fold-change, p-value, and 95% CI (if available) for each gene comparison.

Response: We appreciate the reviewer's suggestions. Although the supplementary tables with specific p-values for each gene expression analysis were incorporated for this new revision, we decided to mention their inclusion more explicitly here, as well as the Ct values obtained for each sample, referring to the average of the two technical replicates performed per sample (lines 312-314). This supplementary material provides the fold change, p-value, and maximum and minimum errors, obtained in the different REST outputs, for each comparison performed and gene analyzed. We would also like to mention that the number of randomizations used for this study is 2,000, which corresponds to the default setting of the REST software and the recommendation by Pfaffl et al. (2002).

References:

  • Pfaffl, M. W., Horgan, G. W., & Dempfle, L. (2002). Relative expression software tool (REST©) for group-wise comparison and statistical analysis of relative expression results in real-time PCR. Nucleic Acids Research, 30(9), e36. https://doi.org/10.1093/nar/30.9.e36

6.2 For haptoglobin you used one-way ANOVA: provide assumptions checks (Shapiro-Wilk statistic and p, Levene’s test statistic and p), exact ANOVA table (F, df, p), post-hoc pairwise comparisons with adjusted p-values and effect sizes (mean differences ±95% CI). Also give the Grubbs’ test results and whether any values were excluded. Present the R script or code used for analysis as supplementary material.

Response: We appreciate the reviewer’s comments. The requested information regarding the one-way ANOVA assumptions (Shapiro-Wilk and Levene’s tests), ANOVA table (F, df, p), post-hoc pairwise comparisons with adjusted p-values and effect sizes, as well as the outlier analysis and the R code used, has been compiled and will be provided as supplementary material.

7- Haptoglobin assay validation

7.1 The Turbitest AA Line assay is validated for serum, but you measured fetal cavity fluids. Please provide analytical validation for this matrix: limit of detection (LoD), limit of quantification (LoQ), linearity, spike-and-recovery (matrix recovery), and intra/inter-assay CVs obtained in fetal fluids. If not performed, the authors should either perform these experiments or clearly rewrite the conclusions about diagnostic usefulness as provisional.

Response: We appreciate the reviewer’s insightful comment regarding the analytical validation of the Turbitest AA Line assay for use in fetal cavity fluids. As correctly noted, this assay is validated for bovine serum, and in our study, it was applied to fetal fluids without formal validation for this specific matrix.

The decision to use this assay was based on its routine application in our diagnostic laboratory and its reliable performance in bovine body fluids. However, we acknowledge that its use in fetal cavity fluids constitutes an extension beyond its validated scope.

We agree that analytical validation, including determination of limit of detection (LoD), limit of quantification (LoQ), linearity, spike-and-recovery, and intra/inter-assay coefficients of variations necessary to confirm the assay’s diagnostic accuracy in this matrix. As such validation was not performed in the present study, we have revised the manuscript to clearly state that the diagnostic utility of haptoglobin in fetal fluids should be considered provisional. This clarification has been incorporated into the Discussion (lines 566-569) and Conclusions (lines 586-589) sections.

We thank the reviewer for highlighting this important aspect, which will guide future research efforts aimed at validating biomarkers in non-standard biological samples.

8- Data availability and reproducibility

8.1 The Data Availability Statement says "Data will be available under request." For transparency please deposit anonymized primary data (individual qPCR Ct values, raw haptoglobin values, sample metadata) and analysis scripts (R, REST outputs) in a public repository (e.g., Zenodo, Figshare, Dryad) and provide accession/DOI.

Response: We welcome the reviewer's suggestions. For greater transparency, we include Table S2 in the supplementary materials with the Ct values obtained for each sample, referring to the average of the two technical replicates performed per sample. The supplementary materials also provide information in the remaining tables on the fold change data, p-values, and maximum and minimum errors obtained in the different REST analyses for each comparison performed and gene analyzed.

9- Pathogen diagnostics and thresholds

9.1 Provide the detection limits for the nested ITS-1 PCR for N. caninum (you mention 10² tachyzoites positive control) and the criteria used to consider a sample positive. For Brucella isolates, state how isolates were confirmed (biochemical, PCR, serotyping). For BVDV, describe the antigen test sensitivity and how cross-contamination was ruled out.

Response: We thank the reviewer for requesting clarification regarding the detection limits and positivity criteria of the nested ITS-1 PCR assay used to detect Neospora. The sensitivity of the assay, as established in our laboratory, is 10 tachyzoites. This threshold was determined using tenfold serial dilutions of DNA extracted from 100,000 tachyzoites obtained from cell culture (NC-1 strain), counted using a Neubauer chamber. A sample was considered positive when a clear amplification product of the expected size (279 bp) was observed on a 1.5% agarose gel electrophoresis stained with SYBR Safe, alongside appropriate positive and negative controls. Each PCR run included DNA equivalent to 10² tachyzoites as a positive control, DNA from experimentally infected mice, and no-template controls (PCR-grade water) to rule out contamination. We have added this information to the Materials and Methods section (lines 158-163) to clarify the diagnostic criteria and analytical sensitivity of the assay.

Brucella isolates were confirmed using both conventional microbiological methods and PCR, as detailed in Section 2.2.2 of the manuscript. Specifically, fetal lung, abomasal contents, and placenta (when available) were cultured on Columbia Blood Agar and Skirrow agar, incubated at 37 °C under a 10% CO₂ atmosphere for 7 days. Colonies were identified based on morphology, Gram staining, and standard biochemical tests including catalase, urease, and oxidase activities, following the protocols described by Alton et al. (1988), which are widely recognized as reference procedures in Brucella diagnostics. Additionally, PCR was performed on pure cultures to detect the IS6501 element of Brucella spp. using ISP1 and ISP2 primers, following a modified protocol based on Saunders et al. (2007). This molecular assay provided further confirmation of the identity of the isolates.

Although serotyping was not performed, the isolates were presumptively identified as B. abortus based on host species (cattle), clinical presentation, and the epidemiological context of Argentina, where B. abortus is the predominant species affecting bovines and other Brucella species are not known to circulate in cattle. This approach is consistent with previous diagnostic practices and national surveillance data.

BVDV detection was performed using a direct immunofluorescence assay on Madin-Darby Bovine Kidney (MDBK) cells, following four blind passages of fetal tissue homogenates cultured at 37°C in 5% CO₂. The assay utilized a commercially available FITC-conjugated porcine polyclonal antiserum specific for BVDV (VMRD Inc., USA), which is routinely used in our diagnostic laboratory and has demonstrated reliable performance in bovine samples. To ensure the specificity of the test and rule out cross-contamination, positive and negative controls were included in each assay. MDBK cultures inoculated with the reference NADL strain of BVDV served as positive controls, while mock-inoculated MDBK cultures (sterile medium only) were used as negative controls. Additionally, technical controls consisting of MDBK cultures incubated without antibody (buffer only) were included to exclude autofluorescence or nonspecific signal. All control cultures were processed in parallel with the samples under identical experimental conditions. A sample was considered positive only when characteristic cytoplasmic fluorescence was observed in the test cultures, and the controls met the expected outcomes (positive control with specific fluorescence, negative controls without signal). These procedures were conducted in a biosafety level 2 laboratory, following strict protocols to prevent cross-contamination. Although the analytical sensitivity of the FITC-conjugated antibody was not quantified in this study, the use of a validated commercial reagent and rigorous control conditions ensured high specificity and minimized the risk of false positives.

This information has been expanded in Section 2.2.1, 2.2.2, and 2.2.3 of the manuscript to improve clarity regarding the diagnostic procedures used.

10- Interpretation and wording

10.1 Some conclusions imply pathogenic mechanisms (Th1→Th2 shift). Given small sample size and cross-sectional design, please soften causal language and explicitly state alternative explanations (timing of infection, tissue specificity, autolysis). Add a paragraph on limitations and how they affect interpretation.

Response: We appreciate the reviewer’s thoughtful observation regarding the interpretation of our findings and the need to soften causal language, particularly regarding the suggested Th1→Th2 immune shift.

In response, we have ensured that the manuscript clearly states that our conclusions are based on cross-sectional observations and a limited sample size, and should therefore be interpreted as preliminary and exploratory. These points were emphasized in both the Discussion and Conclusions sections.

The observed cytokine expression patterns are discussed as potentially reflecting gestational age-related immune modulation, while also acknowledging alternative explanations such as timing of infection, tissue-specific immune responses, and variable degrees of autolysis.

A dedicated paragraph outlining the study’s limitations and their impact on interpretation has been included in the Discussion section. Specifically, we acknowledge:

  • The cross-sectional design prevents temporal inference.
  • The small sample size limits statistical power and generalizability.
  • Cytokine expression was assessed only in spleen tissue, which may not reflect systemic or placental immune responses.
  • Autolysis, although minimal in selected cases, may still affect RNA integrity and protein levels.
  • Control fetuses were opportunistically collected and may differ in breed or maternal condition.

These limitations were already considered in the interpretation of our findings and underscore the need for longitudinal studies, multi-tissue analysis, and quantitative protein-level assays to validate and expand upon our observations.

We thank the reviewer for highlighting this important aspect, which aligns with the cautious and exploratory tone adopted throughout the manuscript.

Reviewer 2 Report (Previous Reviewer 1)

Comments and Suggestions for Authors

dear editor

the authors have answered my concerns and questions and made changes to the manuscript as requested. it is mu recommendation the manuscript to be accepted for publication

Author Response

No changes were suggested by reviewer 2.

Reviewer 3 Report (New Reviewer)

Comments and Suggestions for Authors

The manuscript addresses an important aspect of bovine reproductive immunology—the immune response to fetuses aborted due to intracellular pathogens and "probably infectious" etiology. The novelty of this exploratory study lies in correlating cytokine profiles with gestation stage and infection status. Although exploratory, the study provides valuable and useful information to the field.

However, several methodological, and structural issues need to be addressed before the study can be considered for publication:

Line 64 – [1,3–6] instead of [1,3,4,5,6]). Please use compressed citation ranges, in line with standard journal formatting.

Lines 144-155 - It would be useful for readers to know the amount of DNA template added per PCR reaction (expressed in ng or µL). This detail is important to ensure comparability between studies.
The authors should indicate the analytical sensitivity of the PCR assay (e.g., the minimum detectable number of tachyzoites or genome equivalents). Reporting the detection limit would strengthen the interpretation of negative results.

Lines 156-162 - The reported incubation time of 7 days may be too short, as some B. abortus strains require up to 14–21 days for primary isolation. Please indicate whether cultures were maintained longer or discuss this limitation.
The identification of putative Brucella colonies is currently described only as “standard procedures.” Please provide more specific information (e.g., colony morphology, biochemical tests, etc.) to ensure clarity and reproducibility.
Working with Brucella poses a zoonotic risk. The biosafety level and protective measures adopted during bacteria culture and handling should be specified to demonstrate compliance with laboratory safety standards.

Lines 164-169 – The authors used MDBK cell cultures, but provided too few details about them. Please specify the origin or supplier of the MDBK cells.
The time interval between the blind passages it is important since this can affect the recovery of virus. Please specify the time interval used between the four blind passages.
Also, it would be useful to indicate whether positive and negative controls were used during virus isolation and indirect fluorescent antibody testing. If so, please describe them briefly.

Lines 174-175 - Please specify the exact composition of the antibiotic supplement used in the ASK medium for Campylobacter isolation. This detail is important since different formulations of Skirrow/ASK agar are available and can influence isolation success.

Lines 179-181 - In the description of Tritrichomonas foetus detection procedure, please specify how frequently the liver broth medium was examined microscopically (e.g., daily for seven days). This information is important for reproducibility, as the frequency of monitoring can affect detection sensitivity.

Lines 185-190 - In the histopathology section, please specify the approximate size or thickness of tissue samples fixed in 10% buffered formalin. This detail is important to ensure adequate fixation and reproducibility, since samples with different size may not fix uniformly.
Also, 4 µm it is enough for section thickness. The decimal is unnecessary in this context.

Line 213 - The statement “only samples within the accepted range for high-quality RNA were processed” is vague. Please define the accepted quality threshold, for example by providing the A260/280 ratio range. This will improve transparency and reproducibility.

Lines 216-217 - Since GAPDH can vary under certain biological conditions, validation is important to justify its use as a reference gene. Please clarify whether GAPDH expression was validated for stability across groups. If not validated, this limitation should be acknowledged.

GAPDH stands for ”glyceraldehyde-3-phosphate dehydrogenase”, I suppose. As a rule, abbreviations should be defined at first mention and then use the short form consequently (with very few exceptions). Please check entire manuscript for consistency in abbreviation utilization.

Lines 235-254 - The authors state that pairwise comparisons were conducted using the ”emmeans package”, but do not specify which adjustment method for multiple comparison was applied. Because multiple cytokine genes and group comparisons were analyzed, multiple statistical tests were performed and the probability of committing a type I error is high. Please clarify whether any correction for multiple comparisons was applied to control for type I error. If no correction was used, please justify this choice and discuss how it may affect interpretation of the results. This correction will improve scientific rigor and reproducibility.
It is also interesting if there were situations where the data did not meet one of the ANOVA model assumptions. What happened with this data? How were they handled? This data was excluded or retained in the analysis?
If the Grubbs test identified outlines what happened with these values? This is not specified in materials and methods section. Clarifying these situations is essential for reproducibility and for assessing whether the exclusion (or inclusion) of these data influenced the results.

Lines 256-268 - Since the main objective of the study is the evaluation of cytokine expression and haptoglobin levels in aborted fetuses, the description of how fetuses were selected and grouped is, in my opinion, a methodological step. At present, this information appears in the Results section. I recommend moving the description of selection criteria and groups formation into the Materials and Methods section, as a dedicated subsection. This will improve clarity, ensure methodological transparency, and allow the Results to focus solely on outcomes. If not agree please justify.

In conclusion, the study has clear strengths, particularly in its integration of molecular, immunological, and pathological approaches. However, several methodological details require clarification. These revisions will substantially improve transparency, reproducibility, and scientific rigor. Once addressed these issues, this work has the potential to make a valuable contribution to the field.

My recommendation - Accept after Minor revision

Author Response

Response to Reviewer 3.

The manuscript addresses an important aspect of bovine reproductive immunology—the immune response to fetuses aborted due to intracellular pathogens and "probably infectious" etiology. The novelty of this exploratory study lies in correlating cytokine profiles with gestation stage and infection status. Although exploratory, the study provides valuable and useful information to the field.

However, several methodological, and structural issues need to be addressed before the study can be considered for publication:

Minor Revisions

Minor

  • Line 64 – [1,3–6] instead of [1,3,4,5,6]). Please use compressed citation ranges, in line with standard journal formatting.

Response: Thank you for your observation. We have revised the citation format in line 64 to use compressed ranges, in accordance with the journal's formatting guidelines. The citation now reads: “[1,3–6]” instead of “[1,3,4,5,6].

  • Lines 144-155 - It would be useful for readers to know the amount of DNA template added per PCR reaction (expressed in ng or µL). This detail is important to ensure comparability between studies.

The authors should indicate the analytical sensitivity of the PCR assay (e.g., the minimum detectable number of tachyzoites or genome equivalents). Reporting the detection limit would strengthen the interpretation of negative results.

Response: We appreciate your valuable suggestion. To improve clarity and reproducibility, we have added the amount of DNA template used per PCR reaction. Specifically, 5 µL of DNA at a concentration of 60 ng/µL was added to each reaction, corresponding to 300 ng of DNA per reaction, following the protocol described by Sánchez-Sánchez et al. and Hecker et al. [26,27].

Additionally, we clarified the analytical sensitivity of the assay. As stated, the detection limit was established at 10 tachyzoites, based on serial dilutions of DNA extracted from 100,000 tachyzoites (NC-1 strain), counted using a Neubauer chamber. This information has been incorporated into the revised manuscript to strengthen the interpretation of negative results.

  1. Sánchez-Sánchez, R., Ferre, I., Re, M., Vázquez, P., Ferrer, L. M., Blanco-Murcia, J., Regidor-Cerrillo, J., Pizarro Díaz, M., González-Huecas, M., Tabanera, E., et al. Safety and efficacy of the bumped kinase inhibitor BKI-1553 in pregnant sheep experimentally infected with Neospora caninum tachyzoites. Int. J. Parasitol.: Drugs Drug Resist. 2018, 8, 112-124. https://doi.org/10.1016/j.ijpddr.2018.02.003.
  2. Hecker, Y.P., Burucúa, M.M., Fiorani, F., Maldonado Rivera, J.E., Cirone, K.M., Dorsch, M.A., Cheuquepán, F.A., Campero, L.M., Cantón, G.J., Marían, M.S. et al. Reactivation and foetal infection in pregnant heifers infected with Neospora caninum live tachyzoites at prepubertal age. Vaccines, 2022, 10, 1175. https://doi.org/10.3390/vaccines10081175.

  • Lines 156-162 - The reported incubation time of 7 days may be too short, as some abortus strains require up to 14–21 days for primary isolation. Please indicate whether cultures were maintained longer or discuss this limitation.
    The identification of putative Brucella colonies is currently described only as “standard procedures.” Please provide more specific information (e.g., colony morphology, biochemical tests, etc.) to ensure clarity and reproducibility.
    Working with Brucella poses a zoonotic risk. The biosafety level and protective measures adopted during bacteria culture and handling should be specified to demonstrate compliance with laboratory safety standards.

Response: We appreciate the reviewer’s insightful comments. Regarding the incubation time, based on our previous experience with Brucella isolation from bovine fetal samples and supported by published reports (Fiorentino et al., 2008; Cantón et al., 2022), we have consistently found that a 7-day incubation period is sufficient for primary isolation under the conditions used in our laboratory. In these cases, the bacterial load is typically high, which facilitates early detection. We acknowledge that longer incubation periods (up to 14–21 days) may be necessary for samples with low bacterial burden, such as blood cultures. However, for fetal tissues, the 7-day period has proven reliable and effective.

Brucella colonies were identified based on colony morphology, Gram staining, and conventional biochemical tests, including catalase, urease, and oxidase activities, following the protocols described by Alton et al. (1988). This reference is widely recognized as a standard in Brucella diagnostics and has been extensively cited in the literature. For this reason and given that microbiological identification was not the primary focus of our study, we initially referred to these methods as “standard procedures.” Additionally, PCR was performed on pure cultures to detect the IS6501 element of Brucella spp. using ISP1 and ISP2 primers, following a modified protocol based on Saunders et al. (2007). This molecular assay provided further confirmation of the identity of the isolates.

All procedures involving the isolation and handling of Brucella strains were performed in a Class II biological safety cabinet, in compliance with biosafety level 2 standards. Appropriate personal protective equipment was used throughout, including lab coats, gloves, and face protection, to mitigate zoonotic risk and ensure laboratory safety.

  • Lines 164-169 – The authors used MDBK cell cultures, but provided too few details about them. Please specify the origin or supplier of the MDBK cells.
    The time interval between the blind passages it is important since this can affect the recovery of virus. Please specify the time interval used between the four blind passages.
    Also, it would be useful to indicate whether positive and negative controls were used during virus isolation and indirect fluorescent antibody testing. If so, please describe them briefly.

Response: MDBK cells used for BVDV isolation were obtained from the American Type Culture Collection (ATCC) and provided by the Virology Laboratory of the Specialized Veterinary Diagnostic Service (SDVE), INTA Balcarce, Argentina. These cells were cultured under standard conditions at 37°C in a 5% CO₂ atmosphere. For virus isolation, a 10% homogenate of fetal tissues (spleen, brain, and lung) was prepared and inoculated into MDBK cell cultures. The samples underwent four blind passages, each lasting 72 hours, which is a standard interval in our laboratory to allow sufficient time for viral replication and detection. After the final passage, BVDV antigen detection was performed using a direct immunofluorescence assay with a FITC-conjugated porcine polyclonal antiserum specific for BVDV (VMRD Inc., USA).

To ensure the reliability of the results and rule out cross-contamination, positive and negative controls were included in each assay. MDBK cultures inoculated with the reference NADL strain of BVDV served as positive controls, while mock-inoculated MDBK cultures (sterile medium only) were used as negative controls. Additionally, technical controls consisting of MDBK cultures incubated without antibody (buffer only) were included to exclude autofluorescence or nonspecific signal. All control cultures were processed in parallel with the samples under identical experimental conditions. A sample was considered positive only when characteristic cytoplasmic fluorescence was observed in the test cultures, and the controls met the expected outcomes. These procedures were conducted in a biosafety level 2 laboratory, following strict protocols to prevent cross-contamination.

  • Lines 174-175 - Please specify the exact composition of the antibiotic supplement used in the ASK medium for Campylobacter This detail is important since different formulations of Skirrow/ASK agar are available and can influence isolation success.

Response: Thank you for pointing this out. We agree that the composition of the antibiotic supplement is critical for reproducibility. In our study, the ASK medium contained vancomycin, polymixyn B, cycloheximide and trimethoprim. Specifically, we used vancomycin (10 µg/mL), polymyxin B (1.25 IU/mL), (50 µg/mL) and, trimethoprim (5 µg/mL), all purchased from Sigma-Aldrich (St. Louis, MO, USA). We will add this detail to lines 230–232 of the Methods section to clarify the medium composition.

  • Lines 179-181 - In the description of Tritrichomonas foetus detection procedure, please specify how frequently the liver broth medium was examined microscopically (e.g., daily for seven days). This information is important for reproducibility, as the frequency of monitoring can affect detection sensitivity.

Response: We thank the reviewer for this observation. To improve clarity and reproducibility, we have specified in the revised manuscript that the liver broth medium was examined daily under a light microscope for seven consecutive days. This detail has been added to the corresponding section (2.2.4. Other pathogens) to ensure accurate replication of the detection procedure.

  • Lines 185-190 - In the histopathology section, please specify the approximate size or thickness of tissue samples fixed in 10% buffered formalin. This detail is important to ensure adequate fixation and reproducibility, since samples with different size may not fix uniformly.

Also, 4 µm it is enough for section thickness. The decimal is unnecessary in this context.

Response: We appreciate the reviewer’s observation. To improve clarity and reproducibility, we have specified that tissue samples were approximately 0.5–1 cm thick to ensure adequate fixation. Additionally, the section thickness was corrected to “4 µm” as suggested.

  • Line 213 - The statement “only samples within the accepted range for high-quality RNA were processed” is vague. Please define the accepted quality threshold, for example by providing the A260/280 ratio range. This will improve transparency and reproducibility.

Response: We appreciate the reviewer's comment and have incorporated the requested information in lines 272-274 of the manuscript. The accepted quality threshold was set at the A260/280 ratio range of 1.6 to 2.

  • Lines 216-217 - Since GAPDH can vary under certain biological conditions, validation is important to justify its use as a reference gene. Please clarify whether GAPDH expression was validated for stability across groups. If not validated, this limitation should be acknowledged.

GAPDH stands for ”glyceraldehyde-3-phosphate dehydrogenase”, I suppose. As a rule, abbreviations should be defined at first mention and then use the short form consequently (with very few exceptions). Please check entire manuscript for consistency in abbreviation utilization.

Response: We thank the reviewer for their insightful observations. We acknowledge the importance of using multiple reference genes to improve normalization accuracy in gene expression studies. While GAPDH was selected based on its stable expression in fetal spleen tissue under our experimental conditions, we agree that including additional reference genes would strengthen future analyses. We will incorporate this recommendation in upcoming studies. The clarification of the GAPDH abbreviation was duly incorporated into the manuscript on lines 277–278.

  • Lines 235-254 - The authors state that pairwise comparisons were conducted using the ”emmeans package”, but do not specify which adjustment method for multiple comparison was applied. Because multiple cytokine genes and group comparisons were analyzed, multiple statistical tests were performed and the probability of committing a type I error is high. Please clarify whether any correction for multiple comparisons was applied to control for type I error. If no correction was used, please justify this choice and discuss how it may affect interpretation of the results. This correction will improve scientific rigor and reproducibility.

It is also interesting if there were situations where the data did not meet one of the ANOVA model assumptions. What happened with this data? How were they handled? This data was excluded or retained in the analysis?

If the Grubbs test identified outlines what happened with these values? This is not specified in materials and methods section. Clarifying these situations is essential for reproducibility and for assessing whether the exclusion (or inclusion) of these data influenced the results.

Response: Thank you for this important observation regarding the handling of data in relation to ANOVA assumptions and outlier detection. We agree that clarifying these aspects is essential for reproducibility and transparency.

In our study, the assumptions of the ANOVA model were rigorously evaluated prior to statistical analysis. Specifically, we tested for:

  • Normality of residuals using the Shapiro–Wilk test.
  • Homoscedasticity using Levene’s test and the Breusch–Pagan test.
  • Influential outliers using the studentized residuals test implemented in the outlierTest function of the car package in R.

One data point was identified as an outlier in the haptoglobin dataset (rstudent = –5.96; Bonferroni p < 0.001). This observation was excluded from the main analysis. To assess its impact, we conducted sensitivity analyses by repeating the statistical tests both including and excluding this outlier. The results were consistent, and the exclusion did not alter the overall interpretation of the findings. This procedure is now described in the Materials and Methods section (lines 319–334), and the corresponding R code has been included in the supplementary material.

For the cytokine expression data, statistical analysis was performed using the REST software, which applies a randomization-based approach that does not rely on parametric assumptions. Therefore, conventional outlier detection and normality testing were not applicable to this dataset. However, RNA integrity was assessed using the Ct value of the GAPDH reference gene, and only samples with Ct ≤ 30 were included in the analysis.

We have revised the Materials and Methods section to explicitly describe these procedures and added clarifications to ensure reproducibility. We thank the reviewer for highlighting this important point.

  • Lines 256-268 - Since the main objective of the study is the evaluation of cytokine expression and haptoglobin levels in aborted fetuses, the description of how fetuses were selected and grouped is, in my opinion, a methodological step. At present, this information appears in the Results section. I recommend moving the description of selection criteria and groups formation into the Materials and Methods section, as a dedicated subsection. This will improve clarity, ensure methodological transparency, and allow the Results to focus solely on outcomes. If not agree please justify.

Response: We appreciate the reviewer’s thoughtful suggestion. We agree that the description of fetal selection and group formation is a methodological aspect, and for that reason, it has been thoroughly detailed in the Materials and Methods section (subsection 2.1).

We acknowledge that some of this information is reiterated in the Results section (3.1), particularly regarding the number of cases ultimately included in each group. We consider this repetition purposeful and beneficial for contextualizing the findings and guiding the reader through the interpretation of the results. Since the final composition of the study groups directly influences the outcome analysis, we believe that briefly restating this information in the Results section enhances clarity and narrative flow.

Therefore, we respectfully maintain this structure, as it supports the readability and coherence of the manuscript. Nonetheless, we remain open to making adjustments should the reviewer consider it necessary to improve clarity or alignment with editorial standards.

Reviewer 4 Report (New Reviewer)

Comments and Suggestions for Authors

The manuscript entitled “Cytokine expression and haptoglobin levels in bovine fetuses

spontaneously aborted by intracellular infectious agents and by probable infectious etiology” is well written in clear understandable English and focuses the attention of the reader in changes of bovine fetuses affected by infectious agents.

I am not that who reviewed this paper initially, however I see that it is a revised version of the manuscript. I do not have multiple questions, however there are two Major and few Minor questions which I provide below.

Major

  1. Figure 4 and 5. Control group (non infected fetuses is required, at least 3 fetuses).
  2. Statistics section. The number of analyzed samples is evidently small. Why you not performed analysis using non-parametric tests? E.g. Kruskal-Wallis test? It seems be better in the current case.

Minor

  1. Line 153 instead of 10^2 it will be better to write 100
  2. Line 166 MDBK provide full term
  3. It is a bad idea to use only one reference (Gapdh). For future works you should use multiple references, at least 3.
  4. In table 1 NM numbers of genes it is required to put.
  5. Fig 1 scale bar is absent
  6. Figure 4. What is PIE?
  7. Where the abortuses alive? If yes, how them were killed?
  8. How differed morphology PEI fetuses from other?
  9. Figure 4 and 5 indicate number of biological and technical replicates. The same for other figures with statistics.

Author Response

Response to Reviewer 4.

The manuscript entitled “Cytokine expression and haptoglobin levels in bovine fetuses spontaneously aborted by intracellular infectious agents and by probable infectious etiology” is well written in clear understandable English and focuses the attention of the reader in changes of bovine fetuses affected by infectious agents.

I am not that who reviewed this paper initially, however I see that it is a revised version of the manuscript. I do not have multiple questions, however there are two Major and few Minor questions which I provide below.

Major

  • Figure 4 and 5. Control group (non infected fetuses is required, at least 3 fetuses).

Response: Thank you for your valuable observation. We would like to clarify that control fetuses (non-infected, without inflammatory lesions) matched by gestational age were indeed included in the comparative analyses presented in Figures 3 and 4. These figures show cytokine expression profiles in fetuses aborted due to intracellular agents and probable infectious etiology (PIE), respectively, compared to control fetuses.

However, Figure 5 was specifically designed to compare cytokine expression between fetuses aborted due to confirmed intracellular agents and those with PIE, both at late gestation. This comparison aimed to explore differences in immune profiles between confirmed and suspected infectious causes of abortion. Since the focus of Figure 5 was on the contrast between these two abortion groups, control fetuses were not included in this particular figure to avoid redundancy and maintain clarity.

Nevertheless, we acknowledge the importance of including control data for comprehensive interpretation. To address this, we have ensured that control comparisons are thoroughly presented in Figures 3 and 4, and discussed in the Results and Discussion sections.

  • Statistics section. The number of analyzed samples is evidently small. Why you not performed analysis using non-parametric tests? E.g. Kruskal-Wallis test? It seems be better in the current case.

Response: We appreciate the valuable comment regarding the statistical analysis. The RT-qPCR data in our study were analyzed using REST (Relative Expression Software Tool), which applies the Pair Wise Fixed Reallocation Randomization Test. This approach is specifically designed for relative gene expression data, does not rely on assumptions of normal distribution, and is well suited for small sample sizes. Therefore, REST provides a robust alternative to conventional parametric or non-parametric tests (such as Kruskal–Wallis) for this type of dataset.

Minor

  • Line 153 instead of 10^2 it will be better to write 100

Response: Thank you for your suggestion. We agreed that expressing the quantity as 100 instead of 10² improves clarity and readability. We accepted your recommendation and made the corresponding correction in the manuscript.

  • Line 166 MDBK provide full term

Response: We accepted it and replaced the abbreviation “MDBK” with its full term Madin-Darby Bovine Kidney in the manuscript.

  • It is a bad idea to use only one reference (Gapdh). For future works you should use multiple references, at least 3.

Response: Thank you for your suggestion. We acknowledge the importance of using multiple reference genes to improve normalization accuracy in gene expression studies. While GAPDH was selected based on its stable expression in fetal spleen tissue under our experimental conditions, we agree that including additional reference genes would strengthen future analyses. We will incorporate this recommendation in upcoming studies.

  • In table 1 NM numbers of genes it is required to put.

Response: We appreciate the suggestion to include the NM reference number in Table 1. This suggestion was implemented, and Table 1 now includes the NM number for each gene mentioned, as well as the bibliographic reference.

  • Fig 1 scale bar is absent

Response: Thank you for your observation. We have added appropriate scale bars to each panel in Figure 1 in the revised manuscript.

  • Figure 4. What is PIE?

Response: PIE refers to Probable Infectious Etiology. To improve clarity, we have replaced the phrase “due to probable infectious causes” with the full term followed by the abbreviation: probable infectious etiology (PIE) in the figure legend.

  • Where the abortuses alive? If yes, how them were killed?

Response: All fetuses included in this study were either spontaneously aborted and submitted for routine diagnostic evaluation, or collected postmortem from abattoirs. None of the fetuses were alive at the time of sampling, and no experimental procedures or euthanasia were performed.

  • How differed morphology PEI fetuses from other?

Response: Thank you for your question. The morphological characteristics of fetuses classified under Probable Infectious Etiology (PIE) are detailed in Supplementary Table S1, which includes gestational age, production system, sex, autolysis score, and the most relevant gross and histopathological findings. PIE fetuses exhibited inflammatory lesions such as epicarditis, myocarditis, encephalitis, and interstitial pneumonia, similar to those observed in fetuses infected by confirmed intracellular agents, but without identification of a specific pathogen.

  • Figure 4 and 5 indicate number of biological and technical replicates. The same for other figures with statistics.

Response: We appreciate this observation. In our study, each data point corresponds to an independent biological replicate (an aborted fetus). For each sample, cDNA was analyzed in duplicate (two technical replicates) using the RT-qPCR assay. Statistical analyses in REST were performed using the mean Ct values of these technical replicates for each biological replicate (attached in supplementary material). This information has been clarified in section 2.5 (Cytokine gene expression) of the manuscript and in the figure legends (Figures 4 and 5, as well as in other figures that include this statistical analysis).

Round 2

Reviewer 1 Report (Previous Reviewer 3)

Comments and Suggestions for Authors

My comments have been addressed. Thanks!

Reviewer 4 Report (New Reviewer)

Comments and Suggestions for Authors

Can be accepted.

This manuscript is a resubmission of an earlier submission. The following is a list of the peer review reports and author responses from that submission.

Round 1

Reviewer 1 Report

Comments and Suggestions for Authors

The authors evaluated cytokine expression and hatpoglobin levels in aborted fetuses. The manuscript is well written.

I offer the following questions

1- was there any authorization granted to the authors to be able to use the data?

It seems some of the aborted fetuses were obteined from the veterinary diagnostic service, but no mention that the people submitting the fetuses granted the authorization.
This is a major concern on the ethics of how the data was obtained.

Who covered the cost of the laboratory analysis?

I have some concerns about the fact that there is no authorization reported that granted access to publish this data. The fetuses were submitted by someone to the lab. At this point, who owns the data? do the fetuses become property of the lab? How did the lab gave access to the authors?

2- in the statistical analysis the authors report testing the data for normality of residuals, was there any attempt to identify outliers?

Figures 3,4, and 5 clearly show outliers....

Reviewer 2 Report

Comments and Suggestions for Authors

The manuscript of "Cytokine expression and haptoglobin levels in bovine fetuses spontaneously aborted by intracellular infectious agents and by probable infectious etiology"explore bovine gestational age-dependent immune modulation in response to intracellular pathogen. While the significance of this  study was limited by the small sample size and inproper experiment design, especially there were only two cases of the BVDV and B. abortus infection each, which make it difficult to run the statistical analysis. In addition, the N. caninum,BVDV or B. abortus infection usually cause different clinical outcomes, pathogenesis and cytokine expression,it does not make sense to combine them in a same group based on only gestational age.

Reviewer 3 Report

Comments and Suggestions for Authors

The manuscript addresses an important question but currently suffers from critical design imbalances, insufficient methodological validation, and ethical/data-sharing gaps. Substantive new data (e.g., balanced sampling, RNA integrity metrics, assay validations) or a radical restructuring of the experimental and analytical approach will be required before this work can be considered for publication

  1. Insufficient and imbalanced sample groups

    • The “probable infectious” group lacks mid-gestation fetuses altogether, making any gestational-stage comparison invalid. You must either obtain additional samples to balance all three groups across early, mid-, and late gestation, or substantially revise your conclusions.

  2. Unclear control selection and potential confounders

    • Controls were collected opportunistically at an abattoir and may differ systematically (e.g., breed, health status) from field-diagnosed cases. A more rigorous matching strategy—or at minimum a formal sensitivity analysis—is needed to rule out selection bias.

  1. Lack of RNA quality metrics

    • No RIN or DV200 values are provided. Without demonstrable RNA integrity, the qPCR results are not reliable. You must include these quality-control data or re-run assays.

  2. Single-tissue analysis limits biological interpretation

    • Measuring cytokines only in spleen ignores critical fetal compartments (e.g., placenta, lymph nodes). At minimum, the Discussion must acknowledge that local tissue responses could differ dramatically—and ideally you should provide pilot data from one additional tissue.

  3. Unvalidated assay in fetal fluids

    • The haptoglobin assay’s limit of detection in low-protein fetal fluids is not stated. Validation experiments (e.g., spike-and-recovery) are needed to confirm that the measured concentrations are above assay noise.

  1. Inadequate power and post-hoc control

    • No power calculation is shown for key comparisons. Given small group sizes, the study is likely underpowered—especially for REST’s multiple cytokine tests. You must present a priori power analyses and adjust for multiple testing (e.g., false discovery rate).

  2. Opaque REST parameters

    • The number of randomizations, choice of reference gene(s), and exact p-values are missing. Without these, the fold-change results cannot be independently evaluated.

  1. Missing ethical approval

    • Aborted fetal collection should fall under institutional animal care guidelines. Simply stating “Not applicable” is not acceptable. Provide the ethics committee reference or obtain retroactive approval.

  2. Data deposition required

    • Raw Ct values, haptoglobin readings, and full metadata must be deposited in a public repository prior to publication. “Available on request” does not meet current transparency standards.

  1. Autolysis and mRNA stability

    • Variable post-mortem intervals introduce uncontrolled degradation of mRNA. You need to quantify autolysis scores and correlate these with Ct values—or else limit your claims about absolute cytokine levels.

  2. Developmental regulation confounds

    • The unexpected pattern of higher mid-gestation haptoglobin in controls suggests developmental changes rather than infection effects. You must either include additional gestational age controls or temper your conclusions substantially.